# Oxidative Stress and Human Ovarian Response—From Somatic Ovarian Cells to Oocytes Damage: A Clinical Comprehensive Narrative Review

**DOI:** 10.3390/antiox11071335

**Published:** 2022-07-06

**Authors:** Valentina Immediata, Camilla Ronchetti, Daria Spadaro, Federico Cirillo, Paolo Emanuele Levi-Setti

**Affiliations:** 1Division of Gynecology and Reproductive Medicine, Fertility Center, Department of Gynecology, IRCCS Humanitas Research Hospital, Via Manzoni 56, 20089 Rozzano, Milan, Italy; valentina.immediata@humanitas.it (V.I.); camilla.ronchetti@humanitas.it (C.R.); federico.cirillo@humanitas.it (F.C.); 2Department of Biomedical Sciences, Humanitas University, Via Rita Levi Montalcini 4, 20090 Pieve Emanuele, Milan, Italy; daria.spadaro@humanitas.it

**Keywords:** oxidative stress, oocyte quality, antioxidants, reactive oxygen species, women’s fertility

## Abstract

Basic scientific research on human reproduction and oxidative damage has been extensively performed; however, a more clinical view is still lacking. As a result, exhaustive data on the influence of oxidative stress on human ovarian response and, consequently, on fertility are still lacking. This narrative review aims at summarizing the role of oxidative stress in different conditions associated to female infertility and to list some of the main antioxidant agents. A systematic literature search was performed in May 2022 to retrieve studies regarding the oxidative stress and the human ovarian response from somatic ovarian cells to oocytes damage. Only human studies were included and the authors focused their review, in particular, on clinical implications in order to define a new research perspective on the assessment of any eventual strategy to preserve women’s fertility. Thereby, the authors evaluated the contribution of DNA repair pathways in improving women’s fertility by reducing the DNA damage associated with aging or diseases, such as endometriosis or polycystic ovary syndrome, and eventually, in prolonging the reproductive lifespan after cancer treatment.

## 1. Background and Rationale

Oxidative stress is one of the putative factors involved in the pathologic mechanisms in female infertility [1,2]. However, the role of oxidative stress on human reproduction and, consequently, on pregnancies is difficult to study, mostly because of ethical issues. On the contrary, several animal models on the role of reactive oxygen species (ROS) on female fertility are available in the literature [3,4].

The balance between pro-oxidants, or ROS, and antioxidants ensures the appropriate functioning of most metabolic mechanisms. When this balance shifts toward an excessive generation of ROS, oxidative stress occurs.

Oocyte development is a complex process in which germ and somatic cells are in close association in order to complete the series of ordered maturation and differentiation steps. The oocyte itself can modulate the granulosa cell (GC) differentiation and the follicular development processes. On the other hand, throughout the oocyte growth, granulosa and cumulus cells secrete a wide variety of growth factors that influence the gonadotrophin action in the ovary [5].

Therefore, the ovarian follicle must be considered as a functional unit, where both somatic and germ cells influence the development of an oocyte fully competent to undergo fertilization and embryo formation. If oxidative stress occurs, most ovarian metabolic and endocrine mechanisms may be damaged and women’s fertility impaired.

Oxidative stress is a relevant factor not only for the infertility consequent to reduced oocyte quality or damage to the somatic ovarian cells, but it is also at the basis of several pathological mechanisms responsible for negative pregnancy outcomes and diseases. For example, the pivotal action of oxidative stress in the genesis of preeclampsia has been suggested by different studies [6,7]. The increased generation of ROS due to placental ischemia reperfusion injury appears to play a crucial role in the pathogenesis of this condition [8,9].

Since, in the literature, there is scarce evidence regarding the clinical implications of oxidative stress on the fine human ovarian biology, the present narrative review aims at delving into the present acknowledgment on the topic and thenceforth, at detecting the need for further investigations in the actual world were aging and pollution, as mere examples of causes of cellular oxidative stress, are emerging problems that gynecologists must deal with in order to preserve their patient’s reproductive health.

The present narrative review is, thus, an attempt to guide clinicians through the complexity of physiopathological pathways underlying important female diseases or status in the reproductive age that may limit their fertility due to an impairment in their oocyte quality and ovarian environment.

## 2. Materials and Methods

A systematic literature search was performed in May 2022. The US National Library of Medicine and the National Institutes of Health (NIH) websites were employed to search abstracts having the following key words: “oxidative stress” and “human reproduction”, “female reproduction”, “oocytes”, “follicular fluid”, “infertility”, “ovarian response”. No restrictions for time of publication were set. Only English-written papers were considered. Further relevant papers were found by hand-searching the reference lists of recent articles. Only human studies were included.

## 3. The Importance of Oxidative Balance

ROS are byproducts of cell metabolism, the main ones being superoxide anion (O_2_^•−^), hydroxyl radical (^•^OH), peroxyl (ROO^•^), alkoxyl (RO^•^) and hydroperoxyl (H_2_O_2_). Their constant production results in lipid peroxidation, protein and nucleic acid damage determining strand breaks, loss and deamination of base, together with loss of DNA repair mechanisms and creation of mutations, all inevitably ending in cell death [10]. However, the impact of ROS on proteins is related to the severity of the oxidative stress and the category of proteins involved [11].

In homeostatic conditions, the presence of antioxidant mechanisms allows a balance between ROS production and reduction. Antioxidants are divided into enzymatic, such as the previously cited catalase, superoxide dismutase (SOD) or glutathione transferase, and non-enzymatic, also defined as dietary supplements, which include vitamin C, vitamin E, β-carotene and zinc.

During oxidative respiration, mitochondria convert a small percentage of oxygen into superoxide, which is then converted to hydroxyl radicals. The latter may damage DNA and cause single-strand breaks (SSBs) that can turn to double-strand breaks (DSBs) when multiple lesions occur close together on antiparallel strands. 

Among the different DNA damage types, DSBs are the most detrimental since they imply genomic rearrangements, deletions, translocations and fusions, which cause cellular impairment. The endogenous production of ROS is a major cause of DSBs in somatic [12] but also in germ cells. The spontaneous occurrence of DSBs in oocytes due to the endogenous oxidative stress puts the spotlight on the importance of DNA repair for maintaining oocyte quality [13].

Indeed, in case of oxidant/antioxidant imbalance, unrepaired or incorrectly repaired double-strand breaks (DSBs) may occur [14] and this fact may lead to apoptosis, impaired growth and maturation of the oocyte, infertility [15,16], miscarriages [17,18] or chromosome breaks, translocations, deletions and inversions in the offspring [19,20]. It is well known how chromosomal abnormalities may cause different intellectual and/or physical disabilities, growth retardation and malformations in the offspring, depending on the severity of the chromosomal aberration, or can lead to non-viable embryos.

### 3.1. Follicular Fluid Redox Involvement in Ovarian Follicle Growth and Ovulation

ROS are not only damaging molecules per se, but, if present in normal concentration, they have a key role in the regulation of many signaling pathways essential for female fertility [21].

The ovarian follicle is an example of a system based on complex signaling pathways necessary for the correct maturation of the oocyte and, as a consequence, for proper fertilization and embryo development [22]. A crucial role in this signaling process is played by the follicular fluid, whose microenvironment components comprising proteins, steroids, metabolites, and also reactive oxygen species (ROS), together orchestrate oocyte maturation and ovulation. In particular, the follicular fluid proteome has become a topic of research in recent years. The majority of the follicular fluid protein derives from the extracellular environment and a significant proportion (around 20%) is made up by enzymes: this evidence suggests that the follicular microenvironment is not only influenced by the surrounding environment in which the follicle develops and maturates, but, due to the presence of enzymes, it also acts as a theatre in which fundamental metabolic pathways take place [23].

Through the analysis of the protein profile of human follicular fluid, the identification of a huge number of acute-phase proteins pointed towards the existence of inflammatory processes at the basis of ovulation. Moreover, the presence of different antioxidant enzymes—such as catalase, superoxide dismutase (SOD) or glutathione transferase—also supports the hypothesis that the follicle is able to protect itself against oxidative stress-induced toxic damages [24].

Studies that analyzed the composition of follicular fluid of women under controlled ovarian stimulation highlighted the presence of enzymatic antioxidants, demonstrating the existence of an antioxidant activity that increases proportionally with the increasing size of follicles [25].

Nevertheless, ROS are not considered only damaging agents in the follicular fluid, but, according to some authors, “essential” for ovulation: in particular, hydrogen peroxide seems to have a role in the cumulus oophorus enlargement occurring at the pre-ovulatory LH surge [26]. The presence of ROS in the follicular fluid seems to have a role in follicular growth oocyte maturation and ovarian steroidogenesis [27]. Moreover, oxygen deprivation determined by ROS is a trigger for follicular angiogenesis, which is fundamental for ovarian follicle development and maturation. In addition, ROS are responsible for the process of dominant follicle selection by inducing the apoptosis of the other growing follicles [28,29]. 

Therefore, the follicular fluid, thanks to its antioxidant compounds, is able to prevent the harmful effects of ROS, which, on the other hand, seem to affect the follicular fluid microenvironment not only negatively but also positively by regulating local biological processes.

### 3.2. Defects in the Mitochondrial Respiratory Chain

Mitochondria are fundamental in generating cellular energy, in the form of ATP, from food intake of glucose, which is converted in pyruvate. Producing ATP, mitochondria release ROS, which, if not counterbalanced, can induce oxidative damage to mitochondrial DNA (mtDNA), mutations or deletions. The higher mutation rate in mtDNA compared with nuclear DNA is due to its proximity to ROS generation and the limited repair capacity due to the lack of repair enzymes [30].

As a consequence of cellular aging, mtDNA exposure to ROS leads to an increase in mutations that compromise the effectiveness of this organelle, limiting cellular energy production. This leads to an impaired ability to support cellular functions, for example, chromosomal segregation in the cell division phases. In mature oocytes, which are nonreplicating cells, aging and the associated decrease in mitochondrial functions can be detrimental for fertility in women.

As the largest cell in multicellular organisms, the oocyte contains a high number of mitochondria, since they must guarantee the energy for its growth, maturation, fertilization and embryo formation [31] via oxidative phosphorylation. After fertilization, sperm mitochondria are rapidly degraded, and so embryonic mitochondria are derived exclusively from the oocyte. As previously mentioned, mtDNA has a scarce antioxidant mechanism and no histone protection, which leads to an often-unsuccessful repair system. Therefore, mtDNA is easily damaged by ROS and the quality of oocyte mitochondria, thus, determines the quality of the embryo. The decline in oocyte quality over the years contributes to a maternal age-related decline in women’s fertility [32].

The pathway of the different actors involved in female gonad optimal functionality are graphically shown in Figure 1.

## 4. Oocytes Oxidative Stress and Endometriosis

Endometriosis is a common cause of female infertility; however, the pathological mechanisms responsible for endometriosis-related infertility still remain unclear. Moreover, endometriosis-related infertility is characterized by a lower IVF success rate, and a possible explanation can be identified in a lower quality of oocytes [33]. The cause of worse oocyte quality and, consequently, worse fertility outcomes in this category of women may be oxidative stress. 

ROS generation influences both the endometrial and peritoneal environment, which not only affects fertilization, implantation and embryo development, but also the triggers formation of peritoneal adhesions, in turn, frequently responsible for tubal infertility [34,35].

In endometriosis, erythrocytes and endometrial cells undergoing apoptosis in the retrograde menstrual blood may determine the recruitment of macrophages, which are responsible for ROS release and lipid peroxidation. The resulting oxidative stress leads to an inflammatory reaction, ending up in an increased concentration of cytokines and other pro-inflammatory molecules [36].

Comparing the peritoneal fluid of patients with endometriosis with that of fertile and unexplained infertile women, statistically lower levels of antioxidant agents, such as peroxide dismutase and glutathione peroxidase, were identified in the first group [37].

On the other hand, it was demonstrated that the presence of antioxidant agents, such as dismutase and catalase, significantly decreases pelvic adhesion formation in endometriosis [38].

Furthermore, in endometriosis-affected women, a statistically significant rise in the levels of lipid peroxide was identified [39].

Further, nitric oxide synthase (NOS) may play a role in the pathogenesis of endometriosis-related infertility. In particular, nitric oxide (NO) and NOS, which are normally present in endometrial endothelial glands, are expressed in higher concentrations in this category of women [40,41,42]. This increased expression of NOS in endometriosis results from an excessive generation of free radicals [43,44].

Moreover, in adenomyosis patients, the presence of NOS in the ectopic endometrium persists throughout the menstrual cycle [45].

Therefore, oxidative stress affects fertility by determining an asynchrony between endometrial receptivity and embryo stage, thus, reducing the implantation rate [46].

## 5. Oocytes Oxidative Stress and Polycystic Ovary Syndrome

Polycystic ovary syndrome (PCOS) is one of the most common endocrine disorders that may be diagnosed during the female reproductive period. This syndrome is characterized by the presence of two or more of the following criteria: chronic oligo-ovulation or anovulation, androgen excess and ovaries with polycystic morphology [47]. Typical manifestations are hyperandrogenism, menstrual dysfunction and infertility, but also metabolic disorders, such as dyslipidemia, insulin resistance and impaired glucose tolerance, which can all lead to higher cardiovascular risk and diabetes mellitus [48]. In PCOS patients, an abnormal ovarian extracellular matrix (ECM) composition has been found [49], which can give rise to the above-listed altered hormone secretion. ECM plays an important role in transmitting signals to cells for their proliferation and differentiation. The stage of follicle growth and regression implies a reorganization of the ovarian ECM. In certain conditions of unbalance between the oxidative/antioxidative forces, there can be a shift towards the oxidative status that may affect the matrix metalloproteinases (MMPs) [50] and their activity of cleavage of various ECM structural proteins [51]. Consequently, in a condition of impaired MMP activity, PCOS may develop [52,53]. 

Evidence exists on the role of oxidative stress in PCOS [54,55] and many authors suggested that MMPs may be implicated in the pathogenesis of PCOS through regulating ovarian tissue remodeling [56,57].

Low levels of vitamin D are associated with symptom exacerbation in PCOS; therefore, reduction in vitamin D concentration may be involved in the metabolic and hormonal dysregulation of this clinical condition [58,59].

In addition, a higher prevalence of vitamin D deficiencies among PCOS women was reported in literature [60].

Vitamin D exerts its role on reproduction modulation through acting on vitamin D receptors expressed in GC, influencing the sensitivity to FSH, anti-Mullerian hormone signaling and progesterone synthesis [61,62].

At the level of human ovarian cells, vitamin D not only induces the synthesis of progesterone, but also reduces ROS generation, stimulating the expression of superoxide dismutase (SOD) and catalase [63].

The positive role of exogenous vitamin D administration to infertile PCOS women on follicular differentiation, menstrual regularity and hyperandrogenism reduction was demonstrated by various studies [64,65,66].

Masjedi et al. demonstrated the positive influence of vitamin D in promoting steroidogenesis and antioxidant enzyme activity, and in decreasing ROS production at the level of both normal and PCOS GC. Moreover, their results highlighted, for the first time, the effect of vitamin D supplementation in improving aromatase and 3β-hydroxysteroid dehydrogenase enzyme activities in PCOS GC [67].

## 6. Oocytes Oxidative Stress and Aging

To date, one of the most challenging topics in female reproductive health is the age-related decline in fertility. Over the years, there is a progressive decrease in ovarian reserve, in terms of oocyte number and quality [68,69]. As known, aneuploidies and the impaired growing of mature oocytes [70] are at the basis of the higher rate of infertility or miscarriages in advanced-age women. By contrast, the earliest stages of oocyte development and the causes of the decrease in oocyte quality with aging are less known and debated, even if oocytes exist as primordial follicles for most of women’s lifetime [13].

DNA damage and loss of DNA repair ability contribute to this age-related decline in fertility. Due to this higher DNA repair capability loss with age, oocytes from older women are more prone to exogenous DNA damaging agents, such as radio- or chemotherapy [71,72]. This may be caused by the fact that older women have a smaller ovarian reserve when starting the radio- or chemotherapy [73]. Furthermore, impairment in DNA DSB repair factors is higher in women affected by premature ovarian aging, suggesting a key role for the DNA repair response in fertility. Indeed, if any defect in repair occurs, it may lead to a reduction in oocyte number in the case of apoptosis or to a loss of oocyte quality in case of cell evasion from death activation.

Moreover, mitochondrial defects play a fundamental role in oocyte aging [74]. Studies have shown that the number of mtDNA copies in human oocytes decreases with advancing maternal age [75] and with the decline in ovarian reserve [76]. While oocytes are dormant during most of the woman’s lifespan, they are exposed to harmful endogenous factors, such as ROS and free radicals, which cause their mtDNA to cluster and mutate. In addition, a DNA fragmentation upregulation of cell-free DNA (cfDNA) levels, as a result of apoptotic or necrotic events [77], is found in follicular fluid samples from patients with ovarian reserve disorders. At last, during oocyte formation, a large amount of ATP is required and provided by mitochondria. As mitochondrial dysfunction is associated with oocyte aging [78], this process can be expected to become impaired with age.

## 7. Oocytes Oxidative Stress and Cancer Treatment

Although cancer incidence rates in women in their reproductive age has increased in recent years, mortality is hopefully decreasing due to modern medical improvements [79]. On the other hand, survivors must now cope with the long-term consequences of exposure to cancer therapies.

Both radiotherapy and chemotherapy can cause acute ovarian failure (AOF) that may be transient or permanent. Moreover, they can also cause premature ovarian failure (POF) or premature menopause, during which there is a short return of regular menses with a subsequent loss of ovarian function before the age of 40 years.

Concerning radiotherapy, it can cause ovarian damage, leading to early menopause [80]. GCs are the initial target for radiation damage. When the dividing GCs that support the development of follicles are injured, oocyte maturation may be impaired. This lack of ovulation indicates a loss of female fertility. Just after a few hours of irradiation, cell death can be seen in GCs. With prolonged loss of GCs, the oocyte decreases and loses its viability [81].

The effective sterilizing dose (ESD) of fractionated radiotherapy can cause POF immediately after treatment in 97.5% of patients. ESD decreases with increasing patient age due to the natural decline in oocyte number with age [82]. Doses below the ESD can cause DNA damage that do not result in infertility, but on the contrary, can potentially cause genetic disorders in the offspring, although studies provide little evidence of adverse pregnancy outcomes after exposure [83]. This indicates that DNA repair may occur in oocytes after radiotherapy in order to prevent the inheritance of genetic impairments or that damaged oocytes efficiently degenerate via apoptosis [13].

A disappearance of primordial follicles occurs, occasionally with remnants of degenerating follicles. Collagen takes the place of the ovarian cortical stromal cells and the ovary shrinks. Moreover, radiation injury accelerates the process of small-vessel spontaneous sclerosis and myointimal proliferation to the point of occlusion of the vessel lumen, which is typical in aging [84]. In some women, on the other hand, fertility can be preserved if follicles are relatively radioresistant. This can happen in the late stages of maturation when the GCs are less proliferating. Luckily, in modern times, improved antineoplastic agents and planned radiation therapy may limit the ovarian damage.

Among the chemotherapies, the most toxic to ovaries are the alkylating agents (cyclophosphamide, busulphan and dacarbazine) [84,85,86,87], platinum complexes (cisplatin, carboplatin), taxanes (paclitaxel) and the anthracyclin antibiotic doxorubicin (DXR) [88,89] that can cause DNA breaks, ultimately triggering apoptosis [90], originating firstly in the proliferating GCs of growing follicles [91].

GnRH analogues/agonists (GnRH-a) are considered a measure of protection against ovarian damage during chemotherapy by down-regulating the pituitary gonadal axis and inducing ovarian senescence [92], despite randomized controlled trials still providing conflicting results [90].

It is thought that, thanks to the pituitary gonadotropins that induce a dormant status in the ovary, this may prevent the reaching of the growing phase in which follicles are more sensitive to chemotherapy [93].

## 8. Oocytes Oxidative Stress and Environmental Agents

There are many environmental compounds that act like endocrine disruptors (ED) and interfere with the normal functioning of the endocrine and reproductive systems (e.g., some pesticides, phthalates and phytoestrogens). Importantly, some of these EDs, in particular Bisphenol A (BPA), have been shown to induce DNA damage in both somatic cells [94] and spermatozoa [95]. On the other hand, the direct oocyte impairment caused by DNA damage due to these agents is still not well known.

BPA is an industrial chemical used in the synthesis of polycarbonate plastics and resins contained in food and beverage containers. Since it mimics estrogens and estrogen receptors are expressed in mature oocytes [96], it has been found to increase DNA DSBs and the expression of repair genes [97]. At last, it has been shown to decrease oocyte survival, even in fetal oocytes [98].

Another detrimental environmental toxicant is Benzo(a)pyrene (BaP), a polycyclic aromatic hydrocarbon. BaP is found in cigarette smoke, oils, coal tar, automobile exhaust fumes, margarine, butter, but even in certain foods, such as as fruits, vegetables and cereals. BaP exposure may result in a significant increase in chromosome abnormalities and DNA breaks in oocytes [99].

Ionizing radiation (IR) that may result from the decay of radioactive metals within the Earth [100] in the form of γ-rays and X-rays, is another environmental agent that causes DNA damage within the ovaries [13].

In Table 1 is shown a brief summary of the main mechanisms resulting in oxidative stress in the different analyzed conditions.

## 9. Possible Therapeutic Strategies: Antioxidant Agents

As new primordial follicles cannot be produced after birth, it is essential that the genetic integrity of these follicle oocytes is preserved to ensure optimal ovulation, fertilization and subsequent embryonic development. In protection to oocytes, evolution has evolved a network of DNA damage response pathways, responsible for detecting DNA damage, activating checkpoints leading to cell cycle arrest and coordinating the repair process [101].

Regardless of the fact that many studies in animals confirm the antioxidant role of several agents, such as aloe vera, resveratrol, coenzyme Q10, quercetin, tannic acid and butylparaben, which may be used in animal reproduction to preserve fertility, there is still a lack of evidence in the human counterpart and further investigating studies are needed.

An exception to this statement may be represented by melatonin. Melatonin fulfils its antioxidant action by neutralizing superoxide anion, hydroxyl radicals, hydrogen peroxide, nitric oxide (NO) and peroxynitrite anion [102,103].

For years, melatonin was thought to be only a pineal gland-originated hormone. This concept was called into question with the identification of melatonin in other organs [104]. In particular, ovaries are able to synthetize melatonin at the mitochondrial level: this results in the expression of melatonin in oocytes, cumulus and GCs, where this hormone displays its activity through a series of signaling pathways [105]. 

In this context, melatonin acts by reducing ROS and oxidative stress, promoting oocyte maturation and embryo development [106,107,108].

Different studies reported the effects of melatonin on folliculogenesis, oocyte maturation, steroidogenesis and amelioration of oocyte functioning [103] and quality [109].

A recent study investigating the melatonin signaling pathways involved in human GC metabolism demonstrated that melatonin could act by promoting slower follicular growth and maturation, thus, preventing follicular atresia and early luteinization [110].

## 10. Conclusions

Oocyte quality is particularly sensitive to the metabolic endogenous and exogenous environment and a balanced signaling pathway between oxidant and antioxidant forces is fundamental for a proper follicular maturation.

Women are born with a finite number of oocytes, and it is of key importance that the health of these oocytes is guaranteed throughout their reproductive life to ensure optimal ovulation, fertilization and subsequent embryonic development. The strategy by which the number and quality of oocytes can be preserved is the augmentation of systems capable of rapidly detecting and repairing DNA damage caused by a normal or abnormal metabolic status or by exposure to exogenous agents. Indeed, DNA damage repair is crucial for the integrity of the cellular genome and to its functionality. Noteworthily, for oocytes, it is essential to correct DNA damage in order to prevent the transmission of any genetic mutation to the offspring.

Nevertheless, overall evidence on this topic is primarily based on findings in animal models, while human studies on the ability of oocytes to undertake efficient DNA repair, and the contribution of DNA repair to oocyte quality, are emerging as an essential area of research, though only in recent times. Supporting oocyte DNA repair pathways could be a challenging strategy to protect oocytes from DNA damage and eventual consequent apoptosis in response to endometriosis or PCOS, thus, prolonging the reproductive lifespan after cancer treatment or exposure to environmental agents and toxins. Indeed, female fertility and offspring health rely on the future ability of researchers to provide efficient repair systems to let the oocytes survive from the increasing oxidative stress sources in modern times. Moreover, since a general enhancement in DNA repair systems could impair the efficacy of cancer treatment, a targeting approach would be required in order to preserve both the efficacy of cancer treatment and female fertility.

## Figures and Tables

**Figure 1 antioxidants-11-01335-f001:**
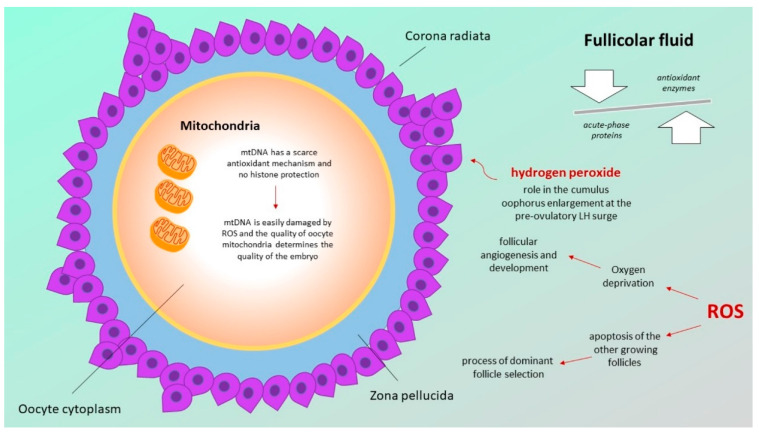
Pathway of the different actors involved in female gonad optimal functionality.

**Table 1 antioxidants-11-01335-t001:** Summary table of the mechanisms resulting in oxidative stress in the different analyzed conditions.

Endometriosis	PCOS	Advanced Age	After Cancer Treatment
▪ Macrophages recruited by apoptotic erythrocytes and endometrial cells are responsible for lipid peroxidation ▪ Increased expression of nitric oxide synthase ▪ Lower levels of peroxide dismutase and glutathione peroxidase in peritoneal fluid	▪ Impaired matrix metalloproteinases activity ▪ Abnormal ovarian extracellular matrix ▪ Low levels of vitamin D	▪ Loss of DNA repair ability ▪ Decreased number of mitochondrial DNA copies ▪ Upregulation of cell-free DNA levels	▪ Radiotherapy: prolonged loss of granulosa cells, accelerated process of small vessel sclerosis and myointimal proliferation ▪ Chemotherapy: DNA breaks, triggering apoptosis

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
