# Peer review of "Oxidative Stress and Human Ovarian Response—From Somatic Ovarian Cells to Oocytes Damage: A Clinical Comprehensive Narrative Review"

_antioxidants, 2022, doi:10.3390/antiox11071335_

Round 1
Reviewer 1 Report
The authors had attempted to explore on the effects of ROS on the human ovaries with concerns on the ovarian cells in particular the oocyte and somatic cells supporting oocyte growth.
However, the review appears to be touching many of the points on the surface and did not delve deep into the unique biology of the human oocyte especially when the focus is as the authors described, "Only human studies were included." Furthermore, the focus and emphasis at the oocyte level seems disjointed as the authors then describe granulosa cells, mitochondria (? oocytes, granulosa cells or ovarian stroma?) etc and how ROS works. It did not focus at the oocyte level, the ovarian follicle level, ovarian stroma etc.
I would suggest to the authors to refocus their review as based on this current version, I am not convinced by the title, "Oxidative Stress and Human Oocyte Quality: A Clinical Comprehensive Narrative Review" as this appears to be focused large on oocyte quantity and quality but I did not find the focus on the oocyte here. It will also be ideal if the authors can do a pictorial diagram linking the pathways together.
Author Response
Author's Notes to Reviewer 1
The authors had attempted to explore on the effects of ROS on the human ovaries with concerns on the ovarian cells in particular the oocyte and somatic cells supporting oocyte growth.
However, the review appears to be touching many of the points on the surface and did not delve deep into the unique biology of the human oocyte especially when the focus is as the authors described, "Only human studies were included."
Thanks to your wise comment, we realized that our work had not been erroneously introduced by an overlook at the physiological biology of the human oocyte growth and the role of its somatic cells, that we added in lines 38-47. The primary intention was to focus the present review only on human studies in order to provide an insight of the influence of oxidation on female human reproduction, specifically on ovarian function, from a clinical point of view, since in the literature there are few publications on this topic. Herein the probable reason why the topic seems touched only on the surface. Henceforth, we tried to extend our review changing the title into “Oxidative stress and human ovarian response: from somatic ovarian cells to oocytes damage. A clinical comprehensive narrative review”, in order to have a wider point of view on the topic. On the other hand, we kept the separate paragraphs in order to deepen the single topics, always guaranteeing a clinical look, only by a human prospective, as it was our primary aim.
Furthermore, the focus and emphasis at the oocyte level seems disjointed as the authors then describe granulosa cells, mitochondria (? oocytes, granulosa cells or ovarian stroma?) etc and how ROS works. It did not focus at the oocyte level, the ovarian follicle level, ovarian stroma etc.
As it is well known, oocyte quality is highly influenced by the ovarian environment, so that it is of basic importance focusing the attention also on the eventual damage that ROS can cause in the ovarian somatic cells when evaluating the ovarian functionality. Hence, we decided to draw a pathway of the different actors (ovarian stroma, granulosa cell, mitochondria, follicle level, oocyte quality) involved in female gonads optimal functionality, as specified graphically in figure 1.
I would suggest to the authors to refocus their review as based on this current version, I am not convinced by the title, "Oxidative Stress and Human Oocyte Quality: A Clinical Comprehensive Narrative Review" as this appears to be focused large on oocyte quantity and quality but I did not find the focus on the oocyte here.
We refocused the review according to your suggestion. As previously said, we modified the title into: “Oxidative stress and human ovarian response: from somatic ovarian cells to oocytes damage. A clinical comprehensive narrative review” thanks to your kind suggestion and we hope that the actual remodeling may help in a better understanding of the topic, since it is our belief that it may help holding a starting point for further research in this field.
It will also be ideal if the authors can do a pictorial diagram linking the pathways together.
As specified above, we added a figure, thanks to your valuable suggestions and a table, according also to the editorial rules.
Author's Notes to Reviewer 2
It seems an interesting manuscript that deals with oxidative stress with oocyte quality.
Thank you for your kind comment.
However, after reading the paper carefully, it looks the abstract and background sections were not written properly. For instance, the abstract and background are same as just the copies.
Thanks to your suggestion, we rewrote the abstract in order that they don’t seem just copies and deal with different topics.
Also, the main topic of this narrative review is dealing with oxidative stress related to oocyte quality other than the mechanisms of female infertility. In fact, the mechanisms of female infertility are involved in many factors, not only oxidative stress related to oocyte quality.
We changed the title into: “Oxidative stress and human ovarian response: from somatic ovarian cells to oocytes damage. A clinical comprehensive narrative review” to change the focus from the single oocyte to the whole ovarian environment, fundamental in female fertility.
Therefore, it is suggested to rewrite the abstract and background and materials carefully.
As previously anticipated, we modified the abstract, deepened the background section and lastly added “infertility” and “ovarian response” as keyword, since we changed the title and the above sections, to give a different point of view of our paper, according to your valuable suggestion.
In summary, this reviewer cannot recommend this manuscript to be accepted for publication with current format and presentation.
We hope that, changing the main topic and focusing the review on the whole human ovarian response to oxidative stress and not only on the oocyte quality, it may make the present work of any help for future researchers, since it opens to the fact that in the literature there are few publications on this topic, especially from a clinical point of view as ours.

Reviewer 2 Report
It seems an interesting manuscript that deals with oxidative stress with oocyte quality. However, after reading the paper carefully, it looks the abstract and background sections were not written properly. For instance, the abstract and background are same as just the copies. Also, the main topic of this narrative review is dealing with oxidative stress related to oocyte quality other than the mechanisms of female infertility. In fact, the mechanisms of female infertility are involved in many factors, not only oxidative stress related to oocyte quality. Therefore, it is suggested to rewrite the abstract and background and materials carefully. In summary, this reviewer cannot recommend this manuscript to be accepted for publication with current format and presentation.
Author Response
Author's Notes to Reviewer 2
It seems an interesting manuscript that deals with oxidative stress with oocyte quality.
Thank you for your kind comment.
However, after reading the paper carefully, it looks the abstract and background sections were not written properly. For instance, the abstract and background are same as just the copies.
Thanks to your suggestion, we rewrote the abstract in order that they don’t seem just copies and deal with different topics.
Also, the main topic of this narrative review is dealing with oxidative stress related to oocyte quality other than the mechanisms of female infertility. In fact, the mechanisms of female infertility are involved in many factors, not only oxidative stress related to oocyte quality.
We changed the title into: “Oxidative stress and human ovarian response: from somatic ovarian cells to oocytes damage. A clinical comprehensive narrative review” to change the focus from the single oocyte to the whole ovarian environment, fundamental in female fertility.
Therefore, it is suggested to rewrite the abstract and background and materials carefully.
As previously anticipated, we modified the abstract, deepened the background section and lastly added “infertility” and “ovarian response” as keyword, since we changed the title and the above sections, to give a different point of view of our paper, according to your valuable suggestion.
In summary, this reviewer cannot recommend this manuscript to be accepted for publication with current format and presentation.
We hope that, changing the main topic and focusing the review on the whole human ovarian response to oxidative stress and not only on the oocyte quality, it may make the present work of any help for future researchers, since it opens to the fact that in the literature there are few publications on this topic, especially from a clinical point of view as ours.

Round 2
Reviewer 1 Report
The authors have made major revisions to the manuscript. I am satisfied with the summary table on the mechanisms and then added pictorial representation makes it easier to follow through the manuscript however, more of the manuscript information can be surmised in this figure and a check on the spelling and formatting will be necessary.
Moderate editing of the English throughout the manuscript will ensure clarity of expression.
Author Response
Thank you very much for recognising our efforts. We tried our best to clarify the authors’ intent throughout the text and to help the reading by providing a graphical representation with both a table and a picture. A wider insight has been added to the picture in order to better summarize the manuscript information. We checked the spelling and formatting and reviewed the English, as kindly suggested.
Reviewer 2 Report
The abstract can be written better than what the authors presented. For instance, the first sentence is not necessary in the abstract.
Author Response
Thank you for your kind suggestions. We removed the first sentence of the abstract and rephrased it in order to improve the reading.